**www.cambridge.org/qrd**

# On the osmotic pressure of cells

Håkan Wennerström[1] and Mikael Oliveberg[2]* ⓘD

[1]Division of Physical Chemistry, Department of Chemistry, Lund University, P.O. Box 124, 22100 Lund, Sweden and
[2]Department of Biochemistry and Biophysics, Arrhenius Laboratories for Natural Sciences, Stockholm University, 106 91 Stockholm, Sweden

## Report

chemical potential of water; cellular osmoticpressure; cellular electrostatic interactions; cellular crowding; functionaladaptation; halophiles

**Author for correspondence:**
*Mikael Oliveberg,
E-mail: mikael.oliveberg@dbb.su.se

### Abstract

The chemical potential of water ($\mu_{H_2O}$) provides an essential thermodynamic characterization of the environment of living organisms, and it is of equal significance as the temperature. For cells, $\mu_{H_2O}$ is conventionally expressed in terms of the osmotic pressure ($\pi_{osm}$). We have previously suggested that the main contribution to the intracellular $\pi_{osm}$ of the bacterium *E. coli* is from soluble negatively-charged proteins and their counter-ions. Here, we expand on this analysis by examining how evolutionary divergent cell types cope with the challenge of maintaining $\pi_{osm}$ within viable values. Complex organisms, like mammals, maintain constant internal $\pi_{osm} \approx$ 0.285 osmol, matching that of 0.154 M NaCl. For bacteria it appears that optimal growth conditions are found for similar or slightly higher $\pi_{osm}$ (0.25-0.4 osmol), despite that they represent a much earlier stage in evolution. We argue that this value reflects a general adaptation for optimising metabolic function under crowded intracellular conditions. Environmental $\pi_{osm}$ that differ from this optimum require therefore special measures, as exemplified with gram-positive and gram-negative bacteria. To handle such situations, their membrane encapsulations allow for a compensating turgor pressure that can take both positive and negative values, where positive pressures allow increased frequency of metabolic events through increased intracellular protein concentrations. A remarkable exception to the rule of 0.25-0.4 osmol, is found for halophilic archaea with internal $\pi_{osm} \approx$ 15 osmol. The internal organization of these archaea differs in that they utilize a repulsive electrostatic mechanism operating only in the ionic-liquid regime to avoid aggregation, and that they stand out from other organisms by having no turgor pressure.

## Introduction

Cellular function is critically dependent on the chemical interplay with the surrounding medium. The most basic characterisation of this surrounding medium is through its intensive thermodynamic variables that, at any moment, strive towards the same values outside and inside the cell. For variables related to solute concentrations, full equilibrium with the surrounding is typically never reached because the cell membrane continuously acts to separate the inside- and outside conditions to control various life processes, for example, energy transduction and selective transport of ions and other solutes. For certain other variables, however, equilibrium across the membrane is reached in relatively short timescales. The most important of these are the temperature ($T$) and the chemical potential of water ($\mu_{H_2O}$).

Although heat conduction is somewhat slower across a lipid membrane than in the aqueous medium, the metabolic processes can only create small temperature gradients. Similarly, diffusion of water is somewhat hindered by the cell membrane, but the permeability remains large enough to yield equilibrium with the outside medium except for situations requiring large and rapid flows of water (Agre, 2006).

Because the surrounding $T$ and $\mu_{H_2O}$ often vary, the cells must somehow respond and adapt. Current organisms display a series of strategies to handle this situation, depending on their evolutionary history, morphology and preferred habitats. The most direct exposure to the environment is experienced by single-cell organisms (Fig. 1). With respect to $T$, these cells are bound to follow the surrounding, so that they can only maintain optimal function in relatively narrow temperature ranges. Even so, some single-cell organisms have specialised by adapting their optimum temperature range to a given habitat, including the extremes of close to the freeze- and boiling points of water (Georlette *et al.,* 2003). Alteration of the intracellular water chemical potential, $\mu_{H_2O}^{cell}$, on the other hand, seems less tolerable. The reason is likely that the value of $\mu_{H_2O}^{cell}$ is determined by the cytosolic composition, and to assure competitive function this needs to be kept within certain boundaries (Wennerstrom *et al.,* 2020). Single-cell organisms have thus developed various stress responses to maintain their function relatively

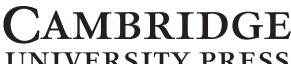

**CAMBRIDGE**
UNIVERSITY PRESS

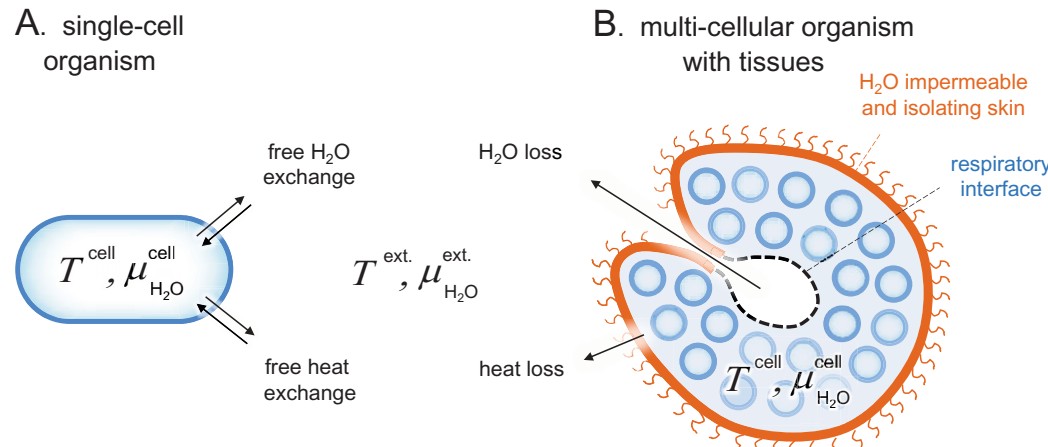

**Figure 1.** Schematic comparison of single-cell and tissue organisms. Sizes are not to scale. (a) The cells of *Archaea, Bacteria* and unicellular *Eukaryota* are directly exposed to their surrounding medium and subject to free $H_2O$ and heat exchange, that is, $T^{cell} \approx T^{ext.}$ and $\mu_{H_2O}^{cell} \approx \mu_{H_2O}^{ext.}$. (b) The cells of higher organisms are typically organised into tissues of specialised function, often including a protective skin. These external skins can present effective barriers for $H_2O$ and heat exchange, allowing the organism to maintain relatively constant internal conditions in a changing environment, that is, $T^{cell} \neq T^{ext.}$ and $\mu_{H_2O}^{cell} \neq \mu_{H_2O}^{ext.}$. A remaining source of $H_2O$ and heat exchange is the respiratory interface in, for example, lungs, gills or insect tracheas.

independent of variations in the external water chemical potential ($\mu_{H_2O}^{ext.}$). These responses include passive exchange of electrolytes, active synthesis of compatible osmolytes, and adaptation of the cell wall to withstand considerable 'hydrostatic' pressures (Wood, 2011; Bremer and Kramer, 2019). In general, however, optimal growth tends to occur at $\mu_{H_2O}^{ext.}$ around that of physiological saline, that is, 150 mM NaCl (Wennerstrom *et al.,* 2020). On the opposite side of the morphologic spectrum are the multicellular organisms that generate their own internal conditions by protecting their cells within a water-impermeable skin and thermally isolating fur or feathers (Fig. 1). Mammals and birds provide the prime example of this strategy, with a strictly regulated body temperature that varies between 32 and 43°C across species (Clarke and Rothery, 2008). Mammals and birds maintain also a strictly regulated value of $\mu_{H_2O}^{cell}$, as apparent to anyone that has suffered thirst. As for the single-cell organisms, the value of $\mu_{H_2O}^{cell}$ matches that of 0.15 M NaCl buffer at around 0.3 osmol (Atkins, 1917). These constant target values for the internal temperature and $\mu_{H_2O}^{cell}$ make the warm-blooded animals not only less sensitive to environmental variations, but also allow them to successfully function in extreme cold where single-cell organisms are bound to be stalled. A striking feature of this organism comparison is that the optimal temperature shows considerable variation, whereas the preferred $\mu_{H_2O}^{cell}$ seems relatively uniform at a value corresponding to 0.15 M NaCl or 0.3 osmol. This value applies also to the majority of multicellular organisms with body temperatures similar to the surrounding, that is, the ectotherms, including reptiles, amphibians, bone- and cartilaginous-fish, arthropods, worms and molluscs. It is thus reasonable to assume that the uniform $\mu_{H_2O}^{cell}$ across evolutionary divergent organisms reflect narrow functional constraints (Record *et al.,* 1998; Wood, 1999; Bolen and Baskakov, 2001; Wood, 2011; van den Berg *et al.,* 2017). However, it is more difficult to see what these constraints are, and how they yield an optimal $\mu_{H_2O}^{cell}$ corresponding to that of physiological saline (Wennerstrom *et al.,* 2020). To shed light on this problem, we discuss here the relation between cell metabolism and water chemical potential from a physicochemical standpoint. Our approach is to first analyse the environmental conditions for

prokaryotic organisms with divergent adaptation strategies, to finally put these in context of some more general conclusions. We arrive at the result that the uniform value of $\mu_{H_2O}^{cell}$ stems from a situation where functional solubility and interactivity of the intracellular components are optimised through their generic intermolecular interactions of largely electrostatic origin. What is often referred to as a non-sustainable value of the intracellular osmotic pressure is actually a non-sustainable electrostatic interplay between the intracellular components.

## Thermodynamic framework: measures of water chemical potential

The chemical potential of water, $\mu_{H_2O}$, is an important thermodynamic property in a variety of contexts. Hence, depending on application, several different ways have been developed to quantitatively account for this property. The most direct is to define the water activity ($a_w$) as

$$\mu_{H_2O} \stackrel{\text{def}}{=} \mu_{H_2O}^{\circ} + kTln(a_w), \tag{1}$$

where $\mu_{H_2O}^{\circ}$ is the chemical potential unity in pure water, where the activity $a_w$ is unity. This definition is analogous to the conventional definition of activities for solutes. It is practical to use Eq. (1) in situations where water is one of several components in similar abundance. The expression in Eq. (1), however, becomes somewhat cumbersome when one is dealing with aqueous solutions where water dominates. For an ideal solution of a solute of concentration $c_s$, the van't Hoff law implies

$$\mu_{H_2O} = \mu_{H_2O}^{\circ} - kTc_s V_w, \tag{2}$$

where $V_w$ is the volume of a water molecule ($\approx 3 \cdot 10^{-29}$ m³). Thus, we have for an ideal solution that the water activity is $a_w = \exp(-c_s V_w)$. A more transparent way to express $\mu_{H_2O}$ is, in this context, to introduce the osmotic pressure ($\pi_{osm}$), which for an ideal aqueous solution is:

$$\pi_{osm} = kTc_s. \tag{3}$$

In textbooks, the osmotic pressure is usually introduced through specifying a measuring procedure. We find it more satisfactory to define the osmotic pressure in terms of the water chemical potential. Thus, we have

$$\pi_{\text{osm}} \overset{\text{def}}{\equiv} \frac{\mu^{\circ}_{H_2O} - \mu_{H_2O}}{V_w}, \tag{4}$$

where the osmotic pressure is a ratio between chemical potential and volume with a dimensionality of energy per volume. It is in general misleading to associate the osmotic pressure with an actual physical pressure. However, energy per volume is dimensionally equivalent to a force per area and there are circumstances when diffusion of water molecules gives rise to an actual pressure difference as, for example, in the conventional textbook definition of the osmotic pressure. An important realisation from the definitions above is that the osmotic pressure is not simply related to the solute concentration as in Eq. (3), unless the solutions are ideal. In live cells, however, the solute concentrations are so high that the solute–solute interactions are substantial, giving a more complex relation between concentration and osmotic pressure. Under such crowded conditions it is common to specify the osmotic pressure by a concentration measure in molar units, the osmolarity $c_{\text{osm}}$, defined as

$$c_{\text{osm}} = 10^{-3}\pi_{\text{osm}}/RT. \tag{5}$$

An advantage of using osmolarity is that it eliminates the explicit temperature dependence for dilute systems. Physiological saline, that is, a solution of 0.154 M NaCl, illustrates the use of this concept. To a good approximation, the $Na^+$ and $Cl^-$ ions dissolve separately in the aqueous medium, which give a total concentration of dissolved species of 0.308 M. Nonetheless, the inherent electrostatic interactions between these species to some extent remain: there is an attraction between anions and cations, and a repulsion between similarly charged ions. In addition, there is a short-range steric repulsion between all dissolved species. At $c_{\text{NaCl}} = 0.154$ M, the contribution from the attraction dominates and this reduces the osmolarity to $c_{\text{osm}} = 0.285$ M, corresponding to an osmotic pressure of 7.25 bar at 310 K. At higher concentrations, the counteracting repulsive term become progressively more important and this increases the osmotic pressure. For example, at 0.5 M NaCl typical for our oceans, the two effects nearly cancel and $c_{\text{osm}} \approx 1.0$ M.

A second biologically relevant medium where it is important to specify the chemical potential of water is in the air or, more generally, in a gas. This is usually accomplished by introducing the partial pressure of water $p_w$ according to

$$\mu_{H_2O} = \mu^{\circ}_{H_2O} + kT\ln\left(\frac{p_w}{p_{\text{sat}}}\right) = \mu^{\circ}_{H_2O} + kT\ln(RH). \tag{6}$$

The first equality of Eq. (6) can in this context be considered as a definition of the partial pressure. It is then only for ideal gases that the partial pressures of the different components add up to the total pressure. The second equality is a definition of the relative humidity (RH), which is commonly used to express the chemical potential of water in the atmosphere. It follows from Eqs. (1) and (6) that the water activity and relative humidity are equal so that $RH = a_w$. A virtue of Eq. (6) is that allows measurements of the water vapour pressure to be used as an accurate method for determining $\mu_{H_2O}$ of a system in equilibrium with a gas phase (Markova et al., 2000). Another possibility to determine $\mu_{H_2O}$ is the osmotic stress technique, provided that care is taken to have a proper reference system

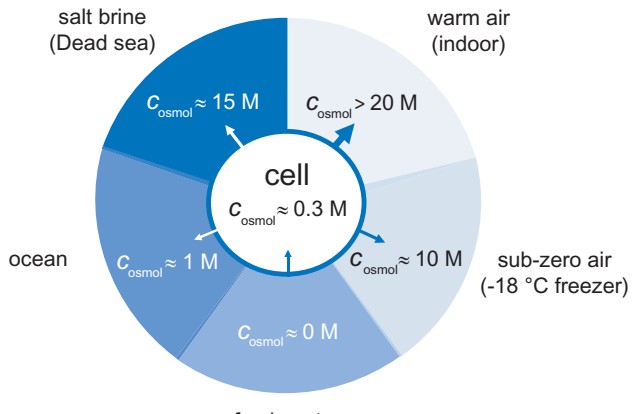

**Figure 2.** External osmotic pressures of different living environments. Arrows show the direction of water flow, where the most desiccating environment is indoor air followed by saturated salt brines.

(Parsegian et al., 2000). The relative humidity of air in equilibrium with physiological saline is 99.49% at 25°C. From a physiological perspective, this shows that relative humidity of around 50% found under normal indoor conditions in temperate zones is, in fact, very desiccating (Fig. 2). Another situation where one reaches very low values for the chemical potential of liquid water is at sub-zero temperatures. Below 0°C, aqueous solutions can only exist at equilibrium if they contain enough solutes to lower their water-chemical potential to that of solid ice; a phenomenon often referred to as freezing-point depression, $\Delta T_{\text{fp}}$. For temperatures close to zero, the depression of the chemical potential of liquid water, $\mu^{\circ}_{H_2O}$, relative to that of ice $\mu^{\circ}_{\text{ice}}$ can be calculated, knowing the heat of fusion of water and

$$\mu^{\circ}_{H_2O} - \mu^{\circ}_{\text{ice}} \approx 2.64k\Delta T_{\text{fp}}. \tag{7}$$

At lower temperatures it is also necessary to account for the difference in heat capacities between liquid water and ice. This more accurate relation is outlined in Table 1, where we also summarise the explicit relations between the different measures of the chemical potential of water. It follows that a physiological-saline solution or the typical cytoplasm of a live cell give just a moderate freezing-point depression of 0.53°C. Thus, if such a cell is put in a household freezer with a temperature of −18°C, it will experience severe desiccation. The ice will typically nucleate extra-cellularly and create a very low water-chemical potential in the environment. As demonstrated above, the usefulness of the different ways of characterising the chemical potential of water is determined by the circumstances. The focus of the current article is on the concentrated aqueous solution inside a live cell. We find then that the osmolarity ($c_{\text{osmol}}$) is the most versatile way of quantifying the chemical potential of water. As pointed out below, we still do encounter situations where cells are exposed to environmental properties that are best described through other measures of the chemical potential of water (Table 1).

## The osmotic pressure of different environments

The availability of water in natural habitats differs quite considerably. One limit is represented by rain- and melt-water. The water is here in essentially pure form, corresponding to a water

**Table 1.** Summary of relations between different measures of the chemical potential of water

| | |
|---|---|
| $\mu_{H_2O} \overset{\text{def}}{=} \mu^{\circ}_{H_2O} + kTln(a_w)$ (per molecule) | Eq. (1) |
| $\mu_{H_2O} \overset{\text{def}}{=} \mu^{\circ}_{H_2O} + RTln(a_w)$ (per mole) | |
| $RH = \left(\frac{p_w}{p_{\text{sat}}}\right) = a_w$ | from Eqs. (1) and (6) |
| $\mu_{H_2O} - \mu^{\circ}_{H_2O} = -N_{Av}V_w \cdot 10^3 RTc_{\text{osm}}$ (per mole) | from Eqs. (4) and (5) |
| $a_w = \exp\left(-N_{Av}V_w \cdot 10^3 c_{\text{osm}}\right)$ | from Eqs. (1), (4) and (5) |
| $\pi_{\text{osm}} = 10^3 RTc_{\text{osm}}$ | from Eq. (5) |
| $\mu_{H_2O}(T) = \mu^{\circ}_{\text{ice}}(T), (T < 273\text{K})$ | |
| $c_{\text{osm}}(T) \approx 0.55(273 - T) - 6 \times 10^{-4}(273 - T)^2, (250\text{K} < T < 273\text{K})$ | from values of water vapour pressure |

activity of unity, zero osmotic pressure and $c_{\text{osmol}} = 0$. In freshwater lakes and rivers the osmotic pressure is slightly higher but still very low. As a river flows towards the sea it takes up more solutes gradually increasing the osmotic pressure, followed by an abrupt change when the water reaches the estuary. In the sea the salt concentration rises to around 0.5 M corresponding to an osmolarity of $c_{\text{osmol}} = 1$ M. Even higher salt concentrations are finally reached below haloclines in pockets at very bottom of the oceans, and in terrestrial salt lakes (Oren, 2002). In these biotopes, the water is typically saturated with salt at several molar, with osmolarities that often exceed 15 M (Fig. 2). For comparison, solid NaCl crystals are in equilibrium with water at a relative humidity of 75% (25°C), corresponding to an osmolarity of $c_{\text{osmol}} = 17$ M. Although not always considered, even higher osmotic pressures challenge the organisms in air. Indoor air in temperate regions, with a temperature around 20°C, typically have a relative humidity in the range 40–60%, corresponding to osmolarities between 20 and 30 M. When the air temperature drops at constant partial pressure of water, the relative humidity increases as $p_{\text{sat}}$ decreases (Eq. (6)). The relative humidity then reaches unity at the dew point, but below 0°C the osmotic pressure increases again because pure water is present as ice. The water chemical potential in a standard household freezer at −18°C yields an osmotic pressure of $c_{\text{osmol}} = 10$ M (Fig. 2). Under these cold conditions, liquid aqueous solutions can only exist if their osmotic pressure is increased, for example by the addition of salt or osmolytes. This is true also for the interior of an unprotected cell.

The evolutionary processes have resulted in a series of adaptation strategies that allow organisms to exist in all these environments. Single-cell organisms like prokaryotes face generally difficulties in environments of very high osmotic pressures, where they have to find ways to resist water loss. Conversely, they have to resist water ingress, excessive swelling and cell-wall rupture when the surrounding osmotic pressure is very low. The significance of the osmotic pressure in regulating bacterial growth is underlined by the fact that, apart from sterilisation, all general food preservation methods rely on creating a high osmotic pressure, be it through drying, freezing or adding salt or sugar. Complex organisms with protective skins or exo-skeletons are better equipped live under high-osmotic pressure conditions, but still require special adaptions to prevent desiccation. One challenge is the need to breathe. With respect to mammals, the air leaving the lungs is practically saturated with water vapour at body temperature. When this air is exchange for ambient air during breathing, there is a substantial water loss (Fig. 1). This loss is compensated by inhaled oxygen that is metabolised to in-body water. Intriguingly at 37°C there is a close balance between the two effects and the net result is that there is a minor loss of water during breathing. Under these conditions, even a 2–5°C increase in body temperature results in an imbalance that makes extra intake of water necessary, as exemplified by the increased need for water during a fever.

## The source of the intracellular osmotic pressure

In a recent paper, we analysed the general features of the interactions between the cellular components of *Escherichia coli* (Wennerstrom *et al.,* 2020). A striking feature of the cellular macromolecules is that the proteins, nucleic acids and lipid membranes all carry net-negative charge (Wennerstrom *et al.,* 2020; Fig. 3). The role of this negative charge is to assure a molecular repulsion that keeps the cytosolic content suitably 'fluid' for function. As proof of principle, the diffusive motions of proteins in live cells display strong correlations with their net-negative surface-charge density, and when they are mutated to obtain a lower repulsive charge they tend to get stuck to their intracellular environment (Mu *et al.,* 2017). This situation, which is observed in both bacterial and mammalian cells (Barbieri *et al.,* 2015; Mu *et al.,* 2017; Ye *et al.,* 2019; Leeb *et al.,* 2020a, 2020b), has led to the interpretation that the diffusive protein–protein interactions are under selective pressure and optimised to assure suitable transient-encounter times (Berg and von Hippel, 1985; Schreiber and Fersht, 1996; Camacho *et al.,* 1999; Mu *et al.,* 2017; Wennerstrom *et al.,* 2020).

When two proteins collide, they will not elastically bounce apart as balls, but will transiently adhere by their close-range interactions (Fig. 3e). The thermal excitations will then repeatedly strive to dislocate the complex but, since most of these are small, the result will for some time be re-binding in slightly new positions. Eventually, there will be a thermal excitation large enough to fully dislocate the proteins, and the process is continued through binding with the next protein in the immediate vicinity. These thermal motions are referred to as Brownian surface diffusion and allow the proteins to search one another's surfaces for putative specific binding sites (Fig. 3f). If the surface-search is too short, the protein can fail to recognise a binding partner, and, if it is too long, the protein will waste time by too extensively searching of non-partners and slow down the cellular machinery. The principal determinants for the protein-interaction potentials and, in turn, the surface-diffusion times, are the close-range

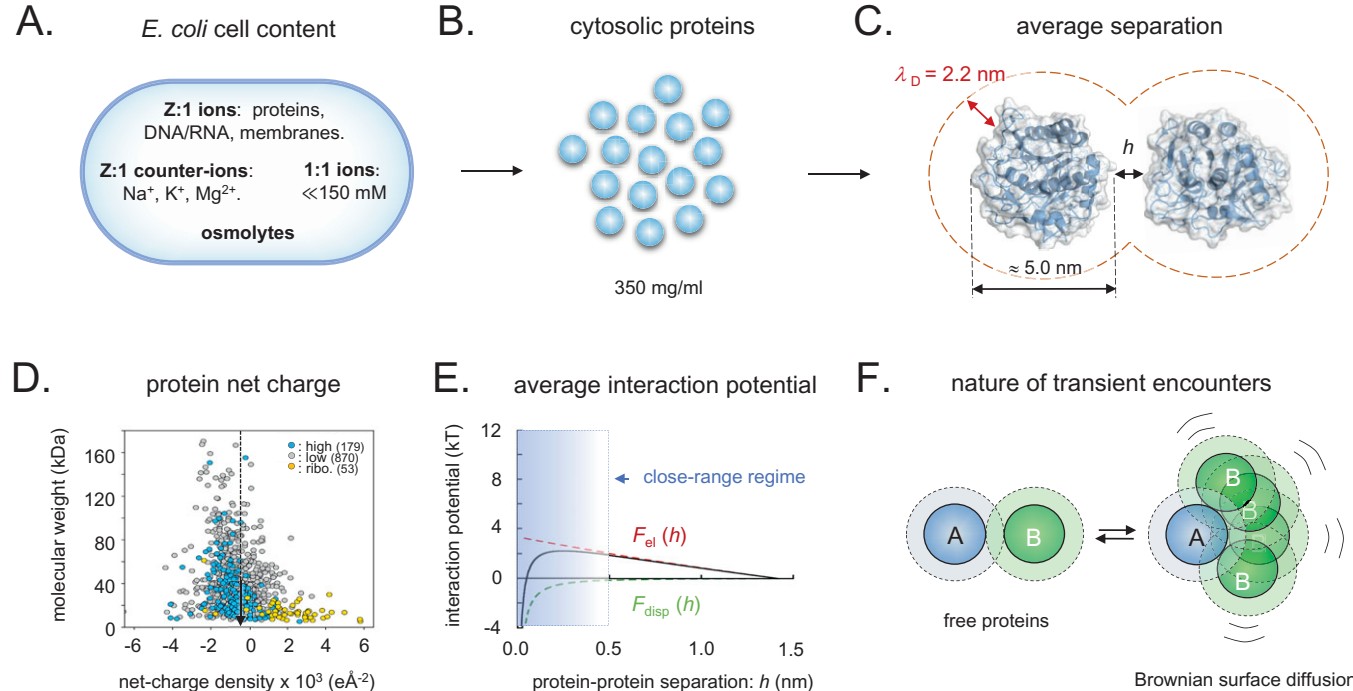

**Figure 3.** Schematic overview of the intracellular conditions in *E. coli* and the length-scales/forces of the protein–protein interactions. (*a*) Ionic content of an *E. coli* cell, where *Z:1 ions* are the net-negatively charged poly-ionic macromolecules, *Z:1 counter-ions* the macromolecules' most abundant positive counter-ions, and *osmolytes* like betaine or trehalose. (*b*) Dimensions of intracellular crowding. (*c*) The separation of average-size proteins is typically less than the estimated intracellular Debye screening length of around 2.2 nm. (*d*) Surface net-charge density *versus* molecular weight for all *E. coli* proteins, showing that the proteome as a whole is biased to net-negative charge. (*e*) Estimated strength of intracellular protein–protein interactions, showing that the electrostatic- and dispersion forces ($F_{el}$ and $F_{disp}$) effectively cancel, facilitating partner search by Brownian surface diffusion. (*f*) Upon random collision, proteins do not elastically 'bounce apart' but 'searches' for putative specific-binding sites by Brownian surface diffusion in the close-range regime, compare, panel E. Dotted radii represent the 2.2 nm screening lengths, compare, panel C.

dispersion force and the protein net charge (Fig. 3*e*). Here, the attractive dispersion force is rather indiscriminate, whereas the latter, typically repulsive, component is under detailed evolutionary control. The effective protein–protein interactions will also depend on the soluble ions of the intracellular medium through their screening of the electric fields, and on the protein concentration, which determine the average protein–protein separation. The concentration of small intracellular anions ($Cl^-$, $HCO_3^-$) appears as relatively low, around 20 mM, compared to a much higher concentration of small cations ($Na^+$, $K^+$) around 150 mM (Wennerstrom *et al.,* 2020). The reason for the mismatch is that most of the intracellular negative charges are carried by the net-negative proteins and other large macromolecules (Mu *et al.,* 2017; Wennerstrom *et al.,* 2020). As such, the cellular cytosol represents a poly-ion system with electrostatic properties that are quite distinct from a 150 mM NaCl solution having the same value of $\mu_{H_2O}$. Most notably, the cytosol is estimated to present a Debye screening length of around 2 nm compared to 0.8 nm in the 150 mM NaCl buffer (Wennerstrom *et al.,* 2020), rendering the long-range electrostatic interactions stronger in the cytosol than in the external growth medium. This complexity calls for an explanation of the source of the cellular osmotic pressure of around 0.3 osmol, which cannot be accounted for by the protein and ion concentrations and Eq. (3). Commonly, the 'missing' osmotic pressure of bacterial cells is attributed to the accumulation of osmolytes in form of compatible solutes (Wood *et al.,* 2001; Wood, 2011, 2015). However, the levels of compatible solutes in unstressed *E. coli* under optimal-growth conditions remain relatively low in the mM regime (Record *et al.,* 1998). As an additional

source of the intracellular osmotic pressure, we have pointed to the soluble proteins themselves (Wennerstrom *et al.,* 2020). Because they occupy around 35% of the available cell volume and also exert significant electrostatic repulsion, their contribution to the osmotic pressure is bound to be substantially higher than estimated from ideal mixing at their concentration of 10–15 mM alone (Wennerstrom *et al.,* 2020). The basic effect of the proteins' repulsion is to restrict their available space, decreasing their translational entropy. Further protein-entropy loss is expected from reduction of the orientational degrees of freedom through the restrictive protein–protein correlations caused by local interaction anisotropies. Proteins are not electrostatically smooth, but show uneven surface-charge decorations that interfere with their tumbling. Both translation and orientation correlations result in strong positive deviations from the ideal-mixing estimate of the osmotic pressure, so that they resist cell-volume reduction by growing progressively stronger as the protein concentration increase. Even so, the magnitudes of these terms remain uncertain and need further exploration. For example, it is indicated from *in-cell* NMR experiments that the Brownian surface diffusion at any moment generates significant levels of transient close-range complexes, that is, dimers and trimers (Leeb *et al.,* 2020*a*), which to some extent decrease the effective protein concentration. The interplay between long-range electrostatic repulsion, medium-range dispersion interaction and short-range direct molecular interactions seem thus to generate complex correlations between translational and orientational degrees of freedom that are yet to be established. Omitting this additional complexity, we estimate the contribution from the

protein–protein interactions to between 25 and 50% of the total intracellular osmotic pressure (SI). Although this estimate remains tentative, it serves to underline that the protein electrostatics plays an intrinsic role in the osmoadaptation of live cells that may also be larger than previously anticipated.

### Functional limitations related to the intracellular water chemical potential

For cells to function efficiently, the network of metabolic processes is dependent on rapid diffusion of the molecular components. As the average diffusion time is proportional to the square of the displacement, it is favourable with short diffusion paths increasing the rate of metabolic transformations. Such short diffusion paths are naturally assured in concentrated systems like the cytoplasm, where the protein–protein separations are similar to the protein sizes (Vazquez, 2010; Dill *et al.*, 2011; Fig. 3*b*). A drawback of this crowded situation, however, is that it is intrinsically sensitive to perturbations. If the protein concentration becomes too low, the diffusion paths will be too long and the metabolic processes will critically slow down. If it becomes too high, the proteins will be arrested by short-range contacts leading to decreased diffusion constants and a seized up cellular machinery (Elowitz *et al.*, 1999; Konopka *et al.*, 2006; Mika *et al.*, 2010; Leeb *et al.*, 2020*b*). In other words, in crowded systems where the range of the repulsive interaction ($\lambda_D$) exceeds the distance to the near neighbours ($h$), any concentration-increase leads to decreased effective repulsion and higher aggregation propensity (Wennerstrom *et al.*, 2020). For *E. coli* the optimum protein concentration seems to be at 30–35% of the available cell volume,

with a matching osmotic pressure in the range of 0.3–0.4 osmol. Transfer of the *E. coli* cells to media with a lower or higher osmotic pressure will hence disturb this optimal protein concentration by swelling or shrinking. This, of course, is a situation that needs to be handled by all types of cells. Existence of a protein concentration optimum for cell metabolism was recently discussed by van den Berg *et al.* (2017), who refers to the phenomenon as crowding homeostasis. However, the intracellular electrostatic interplay is not determined by protein concentration alone, but also by the levels of small ions. If these are increased above the optimal values, the protein–protein repulsion will correspondingly decrease by screening and with similar risk for unspecific aggregation. Accordingly, the protein- and salt concentrations act in concert, and have both to be kept within certain functional limits. The similarity of the basic metabolic processes between different organisms thus explains the narrow range of the optimal osmotic pressure. Higher organisms have protective skins for keeping the internal conditions constant in environments of fluctuating osmotic pressure, but the naked bacteria and archaea have to meet the challenge in a more active way. Accordingly, the risk of drying out or excessively swell presents one of the most ubiquitous stress factors encountered by microorganisms in their habitats (Bremer and Kramer, 2019).

### Prokaryote adaptions to varying external osmotic pressures

#### The basic strategies of osmotic adaptation

Some primitive bacteria like *Acholeplasma* and *Mycoplasma* as well as *Archaea* have only a single lipid plasma membrane as protection against the medium (Gould, 2018; Fig. 4*a*). This makes them

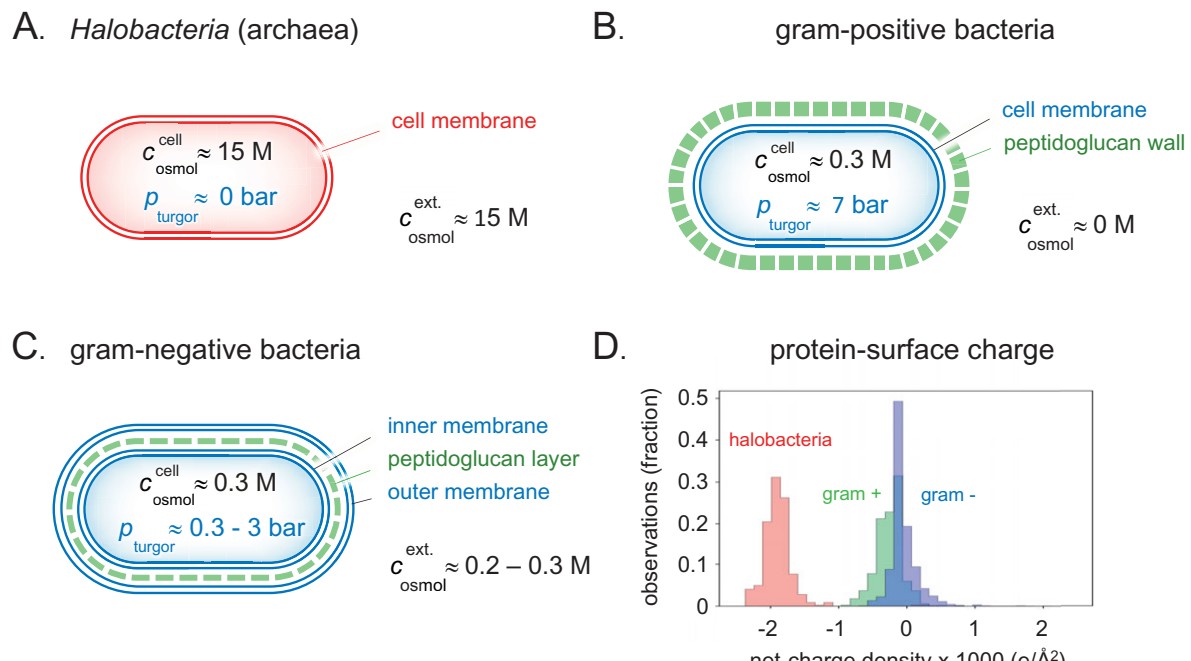

**Figure 4.** Detailed clues to how the different types of prokaryotes handle natural variations in external osmotic pressures are revealed by their cellular architecture. (*a*) Brine-adapted archaea sustain the high osmotic pressure of their environment by a salting-in strategy, that is, they adjust to multi-molar cytosolic salt concentrations and negligible turgor pressure. (*b*) Gram-positive bacteria have adapted a relatively thick external cell wall that can mechanically hold high internal turgor pressures, allowing colonisation of fresh-water biotopes without losing cytosolic content. (*c*) In gram-negative bacteria, the peptidoglucan cell wall is contained in the periplasmic space between the inner- and outer membranes. Although the thickness of the cell wall and ability to high turgor are reduced compared to their gram-positive relatives, this adaptation opens for flexible adjustment to a wide range of external osmotic pressures (Fig. 5). (*d*) The average net-charge densities for the proteomes of the three organism types have evolved to distinct preferred values, reflecting their different lines of physicochemical adaptation.

relatively vulnerable to changes in the external osmotic pressure and they can only thrive under selected stable conditions. At environmental water chemical potentials higher than optimal for these organisms, water loss will rapidly increase the internal protein concentrations to levels where diffusion is severely hindered and unspecific aggregation interferes with the metabolic processes. The most common example of such a desiccating environment is the ocean, with a moderately high water potential of around 1 osmol (Fig. 2). One can identify three possible strategies for a cell to adapt to this situation. The simplest is to allow a higher salt content in the cell. An inevitable consequence of such a measure is that the electrostatic repulsion between the cell components becomes weakened, leading to larger tendency for unspecific aggregation and decreased metabolic rates (Wennerstrom et al., 2020). Given enough time, however, the organism can adapt the genome to produce more negatively charged proteins, but only to a certain level as the electrostatic screening becomes more or less complete at salt concentrations above 0.6 M. This conclusion has a conspicuous exception in brine-adapted Halobacteria, which is discussed in a separate section below. The second possibility for the cell to cope with elevated external water chemical potentials is to import or synthesise compensating osmolytes (Wood et al., 2001; Wood, 2011, 2015). Representative examples of compatible solutes employed by bacteria are proline, glycine betaine, carnitine, proline betaine, dimethylsulfoniopropionate, ectoine/hydroxyectoine, trehalose and glucosylglycerol (Bremer and Kramer, 2019). A benefit of this strategy is that mixtures of water and osmolytes yield a much smaller perturbation of the functional electrostatic interactions than import of 1:1 salts. The third adaptation strategy is basically 'mechanical', where free energy is stored by stretching or bending the cell membrane (Liu et al., 2022). When the cell volume changes as water enters or leaves the cell, the cell membrane has to adjust by stretching or buckling, respectively. There is consequently a contribution to the water chemical potential from this mechanical response. The mechanism is best exemplified for fresh-water environments, where the external osmotic pressure of water is lower than optimal. By simply allowing the water ingress to physically stretch the cell wall, the cell can build up a compensating internal pressure. This pressure difference between the cytosol and the environment is referred to as the turgor pressure, $p_{turgor}$ (Rojas and Huang, 2018). The turgor pressure provides a contribution to $\mu_{H_2O}$ in the cell and, thus, also to the osmotic pressure (Eq. (4)). We can write this intracellular osmotic pressure as a sum $\pi_{cell} = \pi_{mol} + p_{turgor}$. At equilibrium with an external medium of osmotic pressure $\pi_{ext.}$ we have

$$p_{turgor} = \pi_{mol} - \pi_{ext}. \tag{8}$$

In pure melt-water, where $\pi_{ext.}$ is negligible, a cell with an internal molecular osmolarity of 0.3–0.4 osmol attains thus a turgor pressure of 7.5–10 bar. The presence of a turgor pressure leads to a tension or negative lateral pressure, $p_{lat}$, stretching the cell wall. For a spherical cell of radius $R$ with single cell wall of thickness $l_h$, the lateral pressure is given by

$$p_{lat} = -p_{turgor}R/(2l_h). \tag{9}$$

Given the material of the encapsulation, a cell can thus withstand a higher turgor pressure the thicker the protecting wall. For example, the around 30 nm thick peptidoglucan wall of the gram-positive bacterium Bacillus subtilis can resist turgor pressures in excess of 10 bar (Whatmore and Reed, 1990; Misra et al., 2013; Fig. 4b).

## Environmental adaptation 1: the ocean

The ocean has an osmolarity of around 1 M caused by the high NaCl concentration. This is substantially higher than the optimal value for bacteria. The mechanisms for adjusting to an increased osmotic pressure are nicely illustrated by the extensively studied gram-negative E. coli bacterium (Rojas et al., 2014). In an optimal medium of $c_{osmol}^{ext.} = 0.3$ M, E. coli shows a maximum multiplication rate of 1 h and an internal turgor pressure of $\approx$1 bar (Cayley et al., 2000; Fig. 5a). Upon abrupt transfer of the E. coli cells from optimal conditions to a desiccating medium of $c_{osmol}^{ext.} = 1$ M, the first response is shrinkage and stalled growth (Fig. 5a,b). The rapid efflux of water under these hyperosmotic conditions causes the intracellular concentration of proteins and other solutes to increase to physiologically non-sustainable values. In line with the arguments presented above, we consider this state as aggregated and functionally arrested. Following the initial collapse of the cytoplasm, however, the E. coli cells allow import of electrolyte from the medium to restore the cytosolic volume and lowering the protein concentration (Record et al., 1998; Cayley et al., 2000; Wood, 2015). The metabolism now increases to the extent that some growth occurs. Because the increased levels of internal salt excessively screen the electrostatic protein–protein interactions, however, this growth is clearly lower than under optimal conditions. In the next stage of adjustment (Wood, 2011, 2015), the cells synthesise or import osmolytes (Cayley and Record, 2003), combined with expulsion of the transiently imported electrolyte (Cayley et al., 2000; Fig. 5a,b). Although this adjustment brings back the intracellular electrostatic interactions and cellular function to satisfactory levels, it comes with the cost of increased metabolic energy. Because of the leakage of osmolytes across the plasma membrane, the process of keeping the osmolyte concentration stable becomes increasingly energy intensive the higher the osmotic pressure. This means that the E. coli cells cannot sustain growth in high salt concentrations, despite that they have two membrane barriers to reduce the diffusive osmolyte loss.

## Environmental adaptation 2: fresh water

In fresh water the osmotic pressure is very low and bacterial cells experience influx of water during osmotic down-shock. They must respond instantaneously to prevent the membranes from bursting (Fig. 5c). This emergency response involves the opening of various types of mechano-sensitive channels, that act as millisecond-release valves to rapidly dump intracellular solutes into the external medium (Bremer and Kramer, 2019). For the gram-negative bacterium E. coli cells, the result is a turgor-pressure increase that is contained within an acceptable level of around 3 bar, but with the penalty of diluted intracellular content and compromised growth (Fig. 5c). The gram-negative bacteria have double lipid membranes that can take up the turgor pressure (Fig. 6). By regulating the peptidoglucan content in the periplasmic space between the membranes it is possible to divide the pressure drop into two components $p_{turgor}^1$ and $p_{turgor}^2$

$$\begin{aligned} p_{turgor}^1 &= \pi_{mol} - \pi_{mol}^i \\ p_{turgor}^2 &= \pi_{mol}^i - \pi_{ext.}; p_{turgor}(total) = p_{turgor}^1 + p_{turgor}^2, \end{aligned} \tag{10}$$

where $\pi_{mol}^i$ refers to the molecular osmotic pressure in the periplasmic space.

Gram-positive bacteria with thick outer peptidoglucan cell walls (Fig. 6) typically tolerate higher turgor pressures than their

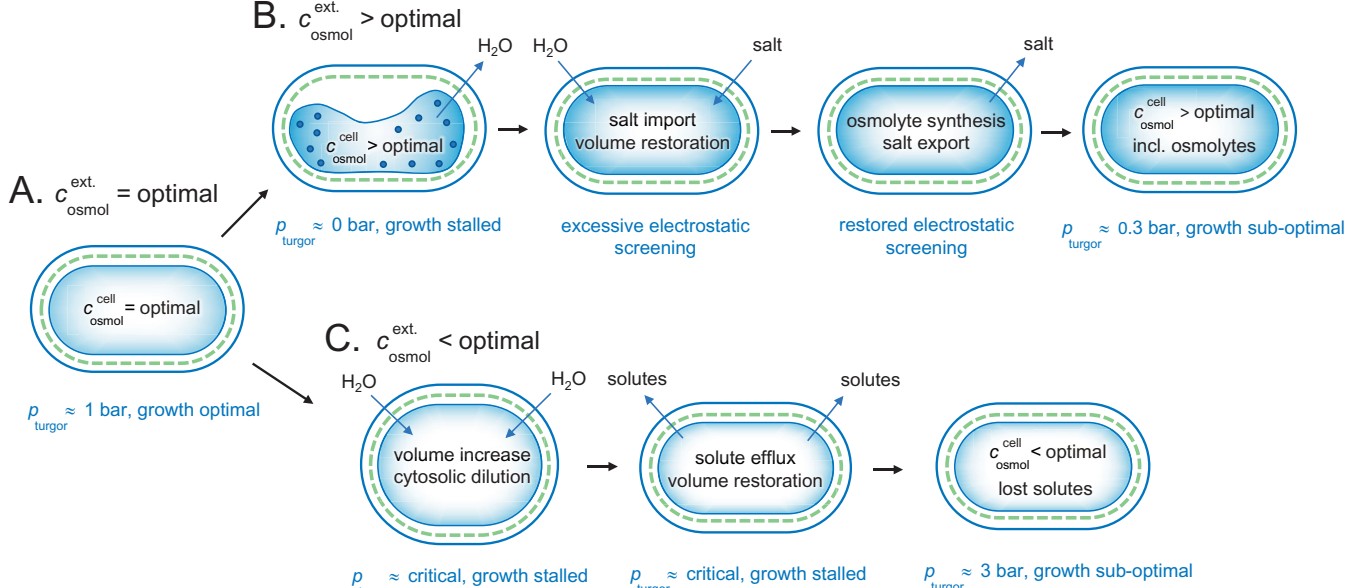

**Figure 5.** Adjustment of a gram-negative bacterium to fluctuating external osmotic pressures. (*a*) Under optimal growing conditions, the cellular and external osmotic pressures are relatively matched and the bacterium maintains a functional turgor around 1 bar. (*b*) Upon osmotic up-shock, for example, external salt increase, the cytosol rapidly collapses due to water loss and growth is stalled. Possibly, this phase involves intracellular aggregation due to excessively increased protein concentrations. Following the collapse, the cell co-imports water and salt to restore volume, but with the drawback of excessive electrostatic screening. This excessive screening is then mitigated more slowly by replacing the salt with osmolytes, leading eventually to an acceptable state with lowered turgor pressure and sustained energetic penalty from maintaining the intracellular osmolyte concentrations constant against a gradient. (*c*) Upon osmotic down-shock, for example, transition into fresh water, the cells are rapidly filled with water and need to expel functional solutes to avoid rupturing. The amount of solute loss is to some extent compensated by increased turgor pressure, but the overall dilution of the cytosol leads to compromised growth. As shown in Fig. 4, the gram-positive bacteria are better equipped to handle this situation as the can hold higher turgor pressures.

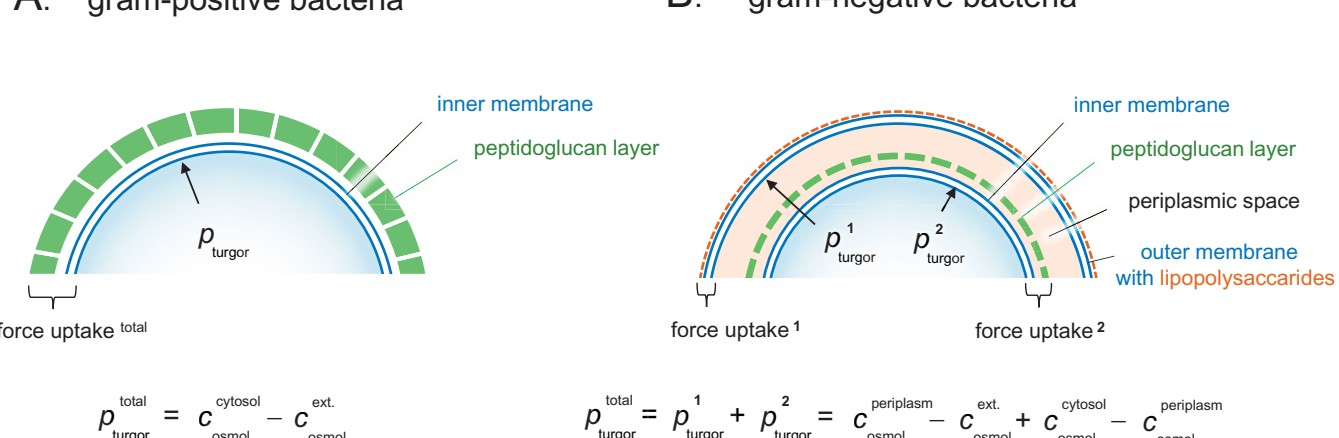

**Figure 6.** Turgor pressure distribution and force uptake in gram-positive *versus* gram-negative bacteria. (*a*) In gram-positive bacteria, the concentration gradient promoting the turgor pressure is maintained by the single inner membrane, whereas the mechanical force is mainly exerted on the outer cell wall. (*b*) In gram-negative bacteria, the turgor pressure can be distributed more flexibly between the cellular envelopes, allowing for a wider range of adjustment strategies, see main text.

gram-negative relatives, and can possibly sustain osmotic down-shock without dumping of cytosolic material and with just marginal effect on the metabolic function. Underlining the selective advantage of this trait, the gram-positive bacteria are found in higher proportions in fresh water than in the sea (Cabello-Yeves and Rodriguez-Valera, 2019). Gram-positive bacteria are also abundant in soil where sudden osmotic down-chock is frequently induced by both rain and flooding. Finally, marine cyanobacteria that sustain lower turgor pressure than their fresh-water relatives, take advantage of the situation by resorting to thinner cell walls

(Walsby, 1986). This trade-off indicates that the high-turgor situation in fresh water is metabolicly costly and cannot easily be avoided by other means, consistent with the notion that adaptation of microorganism across marine- and fresh-water environments is evolutionary demanding (Eiler *et al.*, 2016; Cabello-Yeves and Rodriguez-Valera, 2019). For a comprehensive review of the response of bacteria to osmotic up- and down-shocks, including detailed descriptions of the often mechano-sensitive ion/osmolyte channels controlling cross-membrane transport, see Bremer and Kramer (2019).

## Environmental adaptation 3: the Dead Sea

In the Dead Sea and other salt-lakes as well as in haloclines on the ocean floor, the liquid water is salt saturated resulting in a very high osmotic pressure. Above, we have argued that a generic electrostatic repulsion between cellular components is necessary to prevent aggregation and stalling the metabolic processes. There is a clear limit to which external osmotic pressure most bacteria can withstand, based on the mechanisms discussed above. Salt, and also sugar, in high concentrations yielding osmotic pressures above 2 osmol act as effective food preservatives by hindering bacterial growth. These methods have been around for a very long time, illustrating the fact that there are basic obstacles for bacteria to adjust to such high osmotic pressures even in an evolutionary perspective. Yet there exist living organisms in the Dead Sea. In our previous paper we concluded that the organisation principle of these organisms differs in an important aspect from the common picture. The osmotic balanced is simply maintained by having an internal salt concentration similar to that of the medium. At such salt contents the Debye screening length is negligible and the repulsive electrostatic double layer repulsion present in the cells of other organisms is not operative. Even so, it was recently found (Gebbie et al., 2015; Kjellander, 2016; Smith et al., 2016; Gebbie et al., 2017; Lee et al., 2017; Kjellander, 2018) that at these very high salt contents a different repulsive electrostatic mechanism emerge from ion–ion correlations. However, this is weaker in magnitude so that it requires a major increase of the charge of proteins for the system to function. There are thus major qualitative differences in the genomes of halophiles relative to other bacteria and their proteins have on average a ten times higher charge densities (Gunde-Cimerman et al., 2018; Fig. 4d). In this regime, the screening length increases with increasing concentration (Lee et al., 2017) and there is an eclipse region around 1 M salt where the electrostatic repulsion undergoes a minimum (Wennerstrom et al., 2020). This is also the concentration region where salt acts as a preservative since bacterial growth is hindered by internal aggregation of proteins and other cellular components.

## Conclusions

The chemical potential of water provides an essential characterisation of the environment of living organisms of equal significance as the temperature. We argued that, for most organisms, cellular function is optimised when the molecular component to the osmotic pressure is in the range 0.25–0.4 osmol. This optimum is a compromise between the advantage of having short diffusion paths, whilst the diffusion process is not extensively hindered by unspecific aggregation. At physicochemical level, this balance is maintained by having negatively charged proteins, nucleic acids and lipids in an intracellular medium with low concentration of electrolytes to reduce the screening effect. A main component to the osmotic pressure is thus the soluble proteins and their counterions. We stress here that this close interrelation between protein concentration and the intracellular osmotic pressure means that the two quantities cannot be varied independently. In a growing cell, the increased protein concentration yields an increased osmotic pressure, which in turn can result in a stretching of the membrane. Conceivably, this coupling involves a basic signal for coordinating protein synthesis with lipid synthesis, so that the protein concentration can be kept constant.

Upon exposure to external environments of different osmotic pressures, organisms have to find protective measures. Multicellular organisms, have the ability to maintain an internal water chemical potential close to that of the cellular optimum, which generally corresponds to physiological saline with an osmotic pressure of 0.285 osmol. It is our conclusion that this value is a result of an evolutionary optimization process. The cells of organisms like bacteria and archaea are typically in direct contact with the external medium and respond thus more critically to osmotic pressure changes. The simplest of these unicellular organisms are surrounded by just a single plasma membrane, making them relatively vulnerable and susceptible to thrive only under stable conditions. A major evolutionary step was when gram-positive and gram-negative bacteria developed complex cell wall structures to cope with non-optimal and varying values of the water chemical potential of their environment. These more advanced bacteria can handle low external water chemical potentials by building up an internal turgor pressure. Conversely, when the external water chemical potential exceeds that of the cellular optimum, the bacterial response is to import or synthesise small osmolytes to maintain cell volume, internal protein concentrations as well as electrostatic screening effects within viable limits.

A most remarkable exception to the general rule of optimal cellular osmotic pressures of 0.3–0.4 osmol is found in halophilic arcaea. These organisms multiply in aqueous media saturated in salt, having an osmolarity >10 osmol. To handle this situation their internal electrolyte concentrations, mainly comprising NaCl and KCl, are raised to the levels of ionic liquids. We have previously suggested that these organisms rely on a fundamentally different electrostatic mechanism for maintaining colloidal stability (Wennerstrom et al., 2020), ensuring suitably dynamic protein motions in the cell. Even though the basic metabolic processes in these halophiles are the same as in other cells, the difference in the character of the medium requires a major genomic adaption to alter the molecular properties that govern the protein–protein and protein-nucleic acid interactions. Organism evolution seems thus not only to rely on specific interactions and detailed metabolic pathways, but also on the physicochemical character of the proteomes as a whole. Intriguingly, the latter optimisation involves mainly the hypervariable parts of the protein surfaces that are generally considered non-conserved, pointing to a second layer of evolutionary constraints that remains to be better understood.

**Supplementary Materials.** To view supplementary material for this article, please visit http://doi.org/10.1017/qrd.2022.3.

**Acknowledgements.** We thank Professor Klas Flärd (Lund University), Elloy E. Vallina (Stockholm University) and Jens Danielsson (Stockholm University) for valuable discussions, and the Knut and Alice Wallenberg Foundation (2017-0041) and the Swedish Research Council (2017-01517) for financial support.

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
