## [Reviewer Report]

*Comments to Author*: This report accounts for physicochemical properties of cells making generalized observations focused on osmotic pressure and temperature as universal thermodynamic parameters affecting cell survival in physiological environment, evolutionary adaptation as well as the effect on cellular metabolic function under crowding conditions. This is a nice comprehensive analysis of a topic that is central in cell biology but remains poorly investigated where existing experiments measuring turgor pressure show conflicting results making conclusions vague. Thus, this work is much needed. I therefore recommend publication after minor revision. There are a several suggestions that I would like the authors to consider in their presentation.

1. Osmotic pressure and temperature are coupled through equation of state which authors provide in Eq. 3 for ideal aqueous solutions. Paper provides example of changes in the water chemical potential in the cytoplasm coupled to the freezing point depression. I am curios how the osmotic pressure of thermophilic cells is impacted by extremely high temperatures?

2. I wonder if the reverse process of changes in the intercellular crowding, e.g. during DNA replication in bacterial cells or viral replication in cells, affects the cellular temperature which in turn should have major effect on the metabolic function of cellular enzymes.

3. Authors are discussing the net negative surface charge of cellular components (DNA, proteins and lipids) to prevent aggregation and maintain fluidity. It can be noted however, that in highly crowded environment, negative surface-to-surface charge can create electrostatic friction, having the opposite effect on fluidity. This for instance is observed in tightly packaged DNA phase in bacteriophages, where electrostatic DNA-DNA friction prevents release of viral genome from phage during infection. Could similar scenarios be present in the cell under strong crowding?

4. There is a discussion of optimum molecular crowding effect on cell’s metabolic function. Indeed, tt has been shown that crowding speeds up enzymatic processes (e.g. nuclease digestion of DNA) by bringing enzymes closer to the substrate and reducing diffusion times. In this instance, authors refer to higher aggregation propensity. It is also mentioned in the ms that “the intracellular electrostatic interplay is not determined by protein concentration alone, but also by the levels of small ions.”, where also small ions contribute to the osmotic pressure. Indeed, the concepts of crowding and osmotic pressure are often misinterpreted in the literature and there is a lot of confusion. A study “Osmotic stress, crowding, preferential hydration, and binding: A comparison of perspectives”, by Parsegian et al, PNAS 2000 discusses this. It would be helpful if authors could discuss the differences between crowding and osmotic pressure and also discuss the hydration interactions with the specific examples in cells that they are reviewing.

5. While, authors focus mainly on bacterial cells, it would be interesting to also include in the discussion the effect of tightly condensed DNA on osmotic pressure of cell nuclei in eukaryotic cells. Other interesting scenarios where osmotic pressure plays a central role involve dividing cancer cells as well as rapid viral replication in cells - all strongly affected by crowding.

6. Authors discuss basic mechanical adaptation of cells to the increased osmotic pressure resulting in stretching or bending of the cell membrane. However, recent studies suggest that there are molecular mechanism actively affecting cytoskeleton during mechanical stress, resulting in mechano-protection preventing membrane stretching which could lead to cell rupture. E.g. mechanosensory mechanisms to protect the genome from damage and maintain tissue homeostasis, see Nava et al., 2020, Cell 181, 800-817. This could perhaps be mentioned in this context.

7. Authors are discussing the contribution of the protein-protein pair interaction to the osmotic pressure (with details in SI). This is an interesting aspect. London forces have been shown to promote aggregation of similar charged proteins or DNA under specific solution conditions affected by changes in dielectric permittivity. Is it expected that such pair interactions can lead to aggregation of cellular components under osmotic pressure changes in the cell with specific biological functionality?

---

## [Reviewer Report]

*Comments to Author*: In their Report On the Osmotic Pressure of Cells, Wennerström and Oliveberg provide their biophysical perspective on the roles of molecular interactions, particularly electrostatics, crowding and turgor pressure, in cytoplasmic organization and cellular osmoregulation by prokaryotes.The manuscript summarizes (i) established thermodynamic measures of water potential, (ii) the osmotic pressures of the diverse environments inhabited by prokaryotes, (iii) the origins and implications of the intracellular water potential, and (iv) the mechanisms by which prokaryotes adapt to varying external osmotic pressures.This is useful as most previous reviews on cytoplasmic structure and osmoregulation by prokaryotes, many of which are cited, have been written from microbiological, biochemical and molecular biological perspectives.These authors are particularly interested in item (iii) and outline relevant chemical principles that lead them to a novel perspective.The hypothesis that the physical and chemical mechanisms supporting cytoplasmic structure and function in halophilic archaea differ fundamentally from those operating in other prokaryotes was more fully developed in a previous Perspective by the same authors (reference 1).Therefore this manuscript does not report significant experimental or computational observations of relevance to biological systems, as required for publication in QRB Discovery.

---

## [Reviewer Report]

*Comments to Author*: The manuscript by Wennerström and Oliveberg provides an impressively broad and clear overview of the role of osmotic pressure and water chemical potential in driving cellular processes and how osmotic conditions are maintained.

The opening statement of the abstract is true, underappreciated, and hugely significant. The manuscript is worthy of publication for this statement alone, along with its justification which becomes evident as the remainder of the manuscript unfolds.

The manuscript contains many other illuminating statements and observations relating to electrostatics and osmotics in cells. For example, the authors carefully argue that the cytoplasm is absolutely not iso-osmotic with a 150mM NaCl solution. Despite containing 150mM cations, charge balance is achieved through an order of magnitude lower molar concentration of poly-anions, i.e. the macromolecules of cellular machinery.In this sense, proteins are the “elephant in the room”, or elephants in the cell, huge in volume but (until now) largely ignored in terms of their contribution to the osmotic environment.

The manuscript is impressively broad in its scope, with consideration of surface diffusion, dynamics, screening lengths, halophilic organisms, turgor pressure, and other aspects of intracellular electrostatics. Features are discussed in light of their evolutionary history and intracellular mechanisms. Some of these ideas are not fully developed to the end. But this is not a point of criticism; instead, the result is to lay out plenty of ‘leads’ which I am sure future readers will pick up and study in the details.

The manuscript is beautifully elegant and clear in its exposition, with broad scope and implication. The authors present a new viewpoint regarding cellular driving forces and the evolved intracellular interactions and colligative properties. The thermodynamic framework for understanding and interpreting chemical potential of water and (equivalently) osmolarity is well presented and summarised for non-specialist readers.

Overall, as is evident from the above, I fully support publication of the article. I believe it will influence thinking and open new directions of work along this direction in future.

Some points to consider before publication:

P.5 top, before equation 3: There is an error in the expression for a_w. The exponent should be (-c_s * V_w),not (-c_s / V_w).

P.5 equation 4: here the symbol mu_w is used, different to the mu_H2O used in equations 1 & 2. Do the authors mean to distinguish between liquid water and H2O in general, or should these be the same symbol?

P.7 near top of page, the osmolarity is given the symbol c_osmol; previously it was called c_osm.These should be made consistent.

Table 1, third equation: where did the “18” come from, and why is this no longer dimensionally correct? (We need a factor volume.)

Table 1, fourth equation: as above, the exponent should be dimensionless.

Table 1, last equation: I’m not sure where this comes from, and am not sure it can be correct; it implies that the osmolarity is only a function of temperature, irrespective of solute concentration.

Table 1 general comment: the table should be accompanied by a caption to clarify the definition of terms and their dimensions/units.

p.11, line 10 from top: it is claimed that 150mM NaCl is iso-osmotic with the cytosol (which contains approx. 150mM monovalent cations but mostly poly-valent anions).Isn’t this statement against the definitions of the osmolarity given earlier, since the osmolarity arises from the excess water chemical potential. Of course, the macro-ions (proteins etc) will modify the water chemical potential in a very different way to monovalent ions. Indeed this point is made later in the same paragraph.

P.12, 2nd paragraph, line 3: “…it is favourable with short diffusion paths.”. This is not quite clear; perhaps the authors mean that fast and efficient cellular metabolism is facilitated by close proximity of molecular components?

P.14, eq.8:is pi_mol (internal osmotic pressure) the same as the previously used pi_osm ?If so, could the same symbol be used, and if not could the authors describe how they differ?

P.17 eq.10: what is pi_omg ? In general, it should be checked that all symbols are used consisitently throughtout the manuscript and their definitions are given.

P.4-7 Thermodynamic framework:

The authors have chosen to present a thermodynamically rigorous and clear formulation involving (i) a definition of osmotic pressure as excess chemical potential per molecular volume (eq.4), and (ii) osmolarity as the effective concentration consistent with c proportional to this osmotic pressure (at const. temperature) via the Van’t Hoff law (eq.5). One problem with this approach is that neither c_osm or a_w are defined in terms of c_s, and T, or in terms of a measurable parameter. (In fact, it is correctly emphasised that osmotic pressure should not necessarily be measurable by the traditional text-book method). and therefore, from an experimental point of view, it is not clear how to access this self-consistent framework. I’m aware that these details have been put aside for reasons of clarity, but I think we need an extra few sentences explaining how the excess water chemical potential is accessed experimentally.

---

## [Reviewer Report]

*Comments to Author*: Reviewer #1: In their Report On the Osmotic Pressure of Cells, Wennerström and Oliveberg provide their biophysical perspective on the roles of molecular interactions, particularly electrostatics, crowding and turgor pressure, in cytoplasmic organization and cellular osmoregulation by prokaryotes.The manuscript summarizes (i) established thermodynamic measures of water potential, (ii) the osmotic pressures of the diverse environments inhabited by prokaryotes, (iii) the origins and implications of the intracellular water potential, and (iv) the mechanisms by which prokaryotes adapt to varying external osmotic pressures.This is useful as most previous reviews on cytoplasmic structure and osmoregulation by prokaryotes, many of which are cited, have been written from microbiological, biochemical and molecular biological perspectives.These authors are particularly interested in item (iii) and outline relevant chemical principles that lead them to a novel perspective.The hypothesis that the physical and chemical mechanisms supporting cytoplasmic structure and function in halophilic archaea differ fundamentally from those operating in other prokaryotes was more fully developed in a previous Perspective by the same authors (reference 1).Therefore this manuscript does not report significant experimental or computational observations of relevance to biological systems, as required for publication in QRB Discovery.

Reviewer #2: The manuscript by Wennerström and Oliveberg provides an impressively broad and clear overview of the role of osmotic pressure and water chemical potential in driving cellular processes and how osmotic conditions are maintained.

The opening statement of the abstract is true, underappreciated, and hugely significant. The manuscript is worthy of publication for this statement alone, along with its justification which becomes evident as the remainder of the manuscript unfolds.

The manuscript contains many other illuminating statements and observations relating to electrostatics and osmotics in cells. For example, the authors carefully argue that the cytoplasm is absolutely not iso-osmotic with a 150mM NaCl solution. Despite containing 150mM cations, charge balance is achieved through an order of magnitude lower molar concentration of poly-anions, i.e. the macromolecules of cellular machinery.In this sense, proteins are the “elephant in the room”, or elephants in the cell, huge in volume but (until now) largely ignored in terms of their contribution to the osmotic environment.

The manuscript is impressively broad in its scope, with consideration of surface diffusion, dynamics, screening lengths, halophilic organisms, turgor pressure, and other aspects of intracellular electrostatics. Features are discussed in light of their evolutionary history and intracellular mechanisms. Some of these ideas are not fully developed to the end. But this is not a point of criticism; instead, the result is to lay out plenty of ‘leads’ which I am sure future readers will pick up and study in the details.

The manuscript is beautifully elegant and clear in its exposition, with broad scope and implication. The authors present a new viewpoint regarding cellular driving forces and the evolved intracellular interactions and colligative properties. The thermodynamic framework for understanding and interpreting chemical potential of water and (equivalently) osmolarity is well presented and summarised for non-specialist readers.

Overall, as is evident from the above, I fully support publication of the article. I believe it will influence thinking and open new directions of work along this direction in future.

Some points to consider before publication:

P.5 top, before equation 3: There is an error in the expression for a_w. The exponent should be (-c_s * V_w),not (-c_s / V_w).

P.5 equation 4: here the symbol mu_w is used, different to the mu_H2O used in equations 1 & 2. Do the authors mean to distinguish between liquid water and H2O in general, or should these be the same symbol?

P.7 near top of page, the osmolarity is given the symbol c_osmol; previously it was called c_osm.These should be made consistent.

Table 1, third equation: where did the “18” come from, and why is this no longer dimensionally correct? (We need a factor volume.)

Table 1, fourth equation: as above, the exponent should be dimensionless.

Table 1, last equation: I’m not sure where this comes from, and am not sure it can be correct; it implies that the osmolarity is only a function of temperature, irrespective of solute concentration.

Table 1 general comment: the table should be accompanied by a caption to clarify the definition of terms and their dimensions/units.

p.11, line 10 from top: it is claimed that 150mM NaCl is iso-osmotic with the cytosol (which contains approx. 150mM monovalent cations but mostly poly-valent anions).Isn’t this statement against the definitions of the osmolarity given earlier, since the osmolarity arises from the excess water chemical potential. Of course, the macro-ions (proteins etc) will modify the water chemical potential in a very different way to monovalent ions. Indeed this point is made later in the same paragraph.

P.12, 2nd paragraph, line 3: “…it is favourable with short diffusion paths.”. This is not quite clear; perhaps the authors mean that fast and efficient cellular metabolism is facilitated by close proximity of molecular components?

P.14, eq.8:is pi_mol (internal osmotic pressure) the same as the previously used pi_osm ?If so, could the same symbol be used, and if not could the authors describe how they differ?

P.17 eq.10: what is pi_omg ? In general, it should be checked that all symbols are used consisitently throughtout the manuscript and their definitions are given.

P.4-7 Thermodynamic framework:

The authors have chosen to present a thermodynamically rigorous and clear formulation involving (i) a definition of osmotic pressure as excess chemical potential per molecular volume (eq.4), and (ii) osmolarity as the effective concentration consistent with c proportional to this osmotic pressure (at const. temperature) via the Van’t Hoff law (eq.5). One problem with this approach is that neither c_osm or a_w are defined in terms of c_s, and T, or in terms of a measurable parameter. (In fact, it is correctly emphasised that osmotic pressure should not necessarily be measurable by the traditional text-book method). and therefore, from an experimental point of view, it is not clear how to access this self-consistent framework. I’m aware that these details have been put aside for reasons of clarity, but I think we need an extra few sentences explaining how the excess water chemical potential is accessed experimentally.

Reviewer #3: This report accounts for physicochemical properties of cells making generalized observations focused on osmotic pressure and temperature as universal thermodynamic parameters affecting cell survival in physiological environment, evolutionary adaptation as well as the effect on cellular metabolic function under crowding conditions. This is a nice comprehensive analysis of a topic that is central in cell biology but remains poorly investigated where existing experiments measuring turgor pressure show conflicting results making conclusions vague. Thus, this work is much needed. I therefore recommend publication after minor revision. There are a several suggestions that I would like the authors to consider in their presentation.

1. Osmotic pressure and temperature are coupled through equation of state which authors provide in Eq. 3 for ideal aqueous solutions. Paper provides example of changes in the water chemical potential in the cytoplasm coupled to the freezing point depression. I am curios how the osmotic pressure of thermophilic cells is impacted by extremely high temperatures?

2. I wonder if the reverse process of changes in the intercellular crowding, e.g. during DNA replication in bacterial cells or viral replication in cells, affects the cellular temperature which in turn should have major effect on the metabolic function of cellular enzymes.

3. Authors are discussing the net negative surface charge of cellular components (DNA, proteins and lipids) to prevent aggregation and maintain fluidity. It can be noted however, that in highly crowded environment, negative surface-to-surface charge can create electrostatic friction, having the opposite effect on fluidity. This for instance is observed in tightly packaged DNA phase in bacteriophages, where electrostatic DNA-DNA friction prevents release of viral genome from phage during infection. Could similar scenarios be present in the cell under strong crowding?

4. There is a discussion of optimum molecular crowding effect on cell’s metabolic function. Indeed, tt has been shown that crowding speeds up enzymatic processes (e.g. nuclease digestion of DNA) by bringing enzymes closer to the substrate and reducing diffusion times. In this instance, authors refer to higher aggregation propensity. It is also mentioned in the ms that “the intracellular electrostatic interplay is not determined by protein concentration alone, but also by the levels of small ions.”, where also small ions contribute to the osmotic pressure. Indeed, the concepts of crowding and osmotic pressure are often misinterpreted in the literature and there is a lot of confusion. A study “Osmotic stress, crowding, preferential hydration, and binding: A comparison of perspectives”, by Parsegian et al, PNAS 2000 discusses this. It would be helpful if authors could discuss the differences between crowding and osmotic pressure and also discuss the hydration interactions with the specific examples in cells that they are reviewing.

5. While, authors focus mainly on bacterial cells, it would be interesting to also include in the discussion the effect of tightly condensed DNA on osmotic pressure of cell nuclei in eukaryotic cells. Other interesting scenarios where osmotic pressure plays a central role involve dividing cancer cells as well as rapid viral replication in cells - all strongly affected by crowding.

6. Authors discuss basic mechanical adaptation of cells to the increased osmotic pressure resulting in stretching or bending of the cell membrane. However, recent studies suggest that there are molecular mechanism actively affecting cytoskeleton during mechanical stress, resulting in mechano-protection preventing membrane stretching which could lead to cell rupture. E.g. mechanosensory mechanisms to protect the genome from damage and maintain tissue homeostasis, see Nava et al., 2020, Cell 181, 800-817. This could perhaps be mentioned in this context.

7. Authors are discussing the contribution of the protein-protein pair interaction to the osmotic pressure (with details in SI). This is an interesting aspect. London forces have been shown to promote aggregation of similar charged proteins or DNA under specific solution conditions affected by changes in dielectric permittivity. Is it expected that such pair interactions can lead to aggregation of cellular components under osmotic pressure changes in the cell with specific biological functionality?